# Thirty Years’ History since the Discovery of Pax6: From Central Nervous System Development to Neurodevelopmental Disorders

**DOI:** 10.3390/ijms23116115

**Published:** 2022-05-30

**Authors:** Shohei Ochi, Shyu Manabe, Takako Kikkawa, Noriko Osumi

**Affiliations:** Department of Developmental Neuroscience, Tohoku University Graduate School of Medicine, Sendai 980-8575, Japan; shohei.ochi.a5@tohoku.ac.jp (S.O.); manabe.shu.p6@dc.tohoku.ac.jp (S.M.); kikkawa@med.tohoku.ac.jp (T.K.)

**Keywords:** Pax6, central nervous system, brain patterning, neural stem cells, cell proliferation, neural differentiation, *Sey* mutant, autism spectrum disorder, mouse, rat

## Abstract

Pax6 is a sequence-specific DNA binding transcription factor that positively and negatively regulates transcription and is expressed in multiple cell types in the developing and adult central nervous system (CNS). As indicated by the morphological and functional abnormalities in spontaneous *Pax6* mutant rodents, Pax6 plays pivotal roles in various biological processes in the CNS. At the initial stage of CNS development, Pax6 is responsible for brain patterning along the anteroposterior and dorsoventral axes of the telencephalon. Regarding the anteroposterior axis, Pax6 is expressed inversely to Emx2 and Coup-TF1, and *Pax6* mutant mice exhibit a rostral shift, resulting in an alteration of the size of certain cortical areas. Pax6 and its downstream genes play important roles in balancing the proliferation and differentiation of neural stem cells. The *Pax6* gene was originally identified in mice and humans 30 years ago via genetic analyses of the eye phenotypes. The human *PAX6* gene was discovered in patients who suffer from WAGR syndrome (i.e., Wilms tumor, aniridia, genital ridge defects, mental retardation). Mutations of the human *PAX6* gene have also been reported to be associated with autism spectrum disorder (ASD) and intellectual disability. Rodents that lack the *Pax6* gene exhibit diverse neural phenotypes, which might lead to a better understanding of human pathology and neurodevelopmental disorders. This review describes the expression and function of Pax6 during brain development, and their implications for neuropathology.

## 1. Introduction

It has been 30 years since *Pax6/PAX6* was identified as a gene responsible for congenital anomalies of the eye in mice and humans [1,2]. The gene turned out to be a member of a paired box (Pax) family encoding transcriptional factors that also work in brain development [3]. Regarding structure, Pax proteins share a common DNA-binding domain called the paired domain (PD) [4]. Some members of the Pax family, including Pax6, have a homeodomain (HD), i.e., another DNA-binding domain (Figure 1A) [5,6]. The molecular structure and function of Pax6 are well preserved from *Drosophila* to mammals [7,8]. It has been reported that the PD is necessary for the regulation of embryonic neurogenesis, in which a mutant lacking the HD of Pax6 showed only subtle defects [9]. Therefore, the PD exerts a key role during brain development [9].

Pax6 first garnered attention due to its role in eye development. Spontaneous *Small eye* (*Sey*) mutant mice were identified from the phenotype of microphthalmia as a heterozygous phenotype, while *Pax6* homozygous mutant (*Sey/Sey)* mice completely lacked the formation of the eyes and nose and died soon after birth [10]. The *Pax6* gene mutated in *Sey/Sey* contains a stop codon in the coding region before the homeodomain (Figure 1A) [1]. The *Pax6* gene was also identified in spontaneous mutant mice (*Sey/Sey*) and rats (*rSey2/rSey2*) showing similar ocular phenotypes [11,12,13]. The gene responsible for the *eyeless* mutant in *Drosophila* was found to be a *Pax6* homolog [14]. The ectopic expression of *eyeless* induced the formation of compound eyes outside the head region [7]. Thus, Pax6 is required and sufficient for eye formation.

Further study has revealed that Pax6 is expressed in various regions in the developing central nervous system (CNS) from the initial stage when the neural plate is induced [15]. In the developing CNS, Pax6 is expressed in the telencephalon [16], diencephalon [17,18], rhombomeres [19,20], and spinal cord [20,21,22]. At early stages, Pax6 is expressed in the ventricular zone (VZ), where neural stem cells (NSCs) are located (see “Pax6 in neurogenesis”) [23,24]. At later stages, Pax6 is expressed in the neurons of specific brain regions such as the olfactory bulb [20,25], amygdala [13,26], thalamus [13,27], and cerebellum [13,20,28]. *rSey^2^/rSey^2^* rats lack the olfactory bulb, yet they have an olfactory bulb-like structure at the lateral position in the neocortex [20]. *Sey*/*Sey* mice and *rSey^2^/rSey^2^* rats exhibit a reduction in the size of the forebrain and the cortical thickness (Figure 1B,C) [29].

Outside the CNS, PAX6/Pax6 is expressed in the lens [20,30,31], corneal epithelium [32,33], retinal neuroepithelium [34,35], and olfactory placodes/epithelium [25,36,37]. Pax6 is also involved in the development of other tissues such as the pancreas, pituitary gland, and even in the testes [21,38,39,40] (reviewed in [13,20,37,41,42,43,44]). In this review article, we would like to focus on the role of Pax6 in brain development, taking neurodevelopmental disorders into account.
Figure 1Schematic structure of Pax6 protein and cortical phenotypes in the *Pax6* spontaneous homozygous mutant (*Sey/Sey*) and *Pax6* splice variant (*Pax6(5a)*) mice. (**A**) Schematic structure of the *Pax6* protein in the wild-type (WT), *Sey/Sey,* and *Pax6(5a)* mice. One base-pair substitution in exon 8 of the *Pax6* gene causes a Gly194Stop nonsense mutation, causing a stop codon downstream at the PST site, thereby resulting in the truncated Pax6*^Sey^* protein. The canonical PD binds via its N-terminal PAI domain to the DNA; the insertion of the 14 amino acids from exon 5a into the PAI domain leads to the mutant PD(5a) [45]. PD: paired domain, HD: homeodomain, PST: Pro-Ser-Thr rich region. (**B**) Schematic illustration of the head of WT and *Sey/Sey* mice. Pax6 shows a gradient of expression in the AP axis in WT. *Sey/Sey* mice cause a failure of eye and olfactory bulb formation and rostralization in the telencephalon. OB: olfactory bulb, Tel: telencephalon, M1: motor, S1: sensory, A1: auditory and V1: visual areas. (**C**) Schematic illustration of a WT and *Sey/Sey* mouse telencephalon. *Sey/Sey* mice cause dorsalization in the telencephalon. Cx: cortex, BG: basal ganglia. The graphical diagram has been redrawn from [9,20,46].
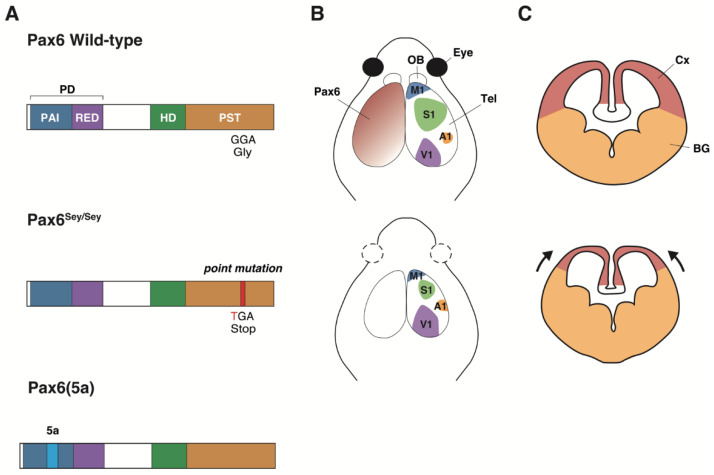


## 2. Pax6 in Cortical Patterning

As mentioned above, Pax6 is considered to be expressed from the initial developmental stage of the CNS, when it plays key roles in the regionalization of the neuroectoderm and neural tube, i.e., the primordium of the CNS [47]. Each region of the CNS is specialized before and during neurogenesis [48]. Although Pax6 is involved in the patterning of the ventral regions of the brain stem and spinal cord [20,49,50], we focus here on the cortical patterning. As described later, the initial patterning can also influence the ontology of neurodevelopmental disorders.

### 2.1. Anteroposterior Axis

The telencephalon is a highly regionalized organ, which is functionally and morphologically diverse. In early brain development, the telencephalon is patterned along the anteroposterior (AP) and dorsoventral (DV) axes, resulting in the formation of various cortical and subcortical areas [51].

This patterning is first attributed to the action of secreted signaling factors such as Wnt, bone morphogenetic protein (BMP), and FGF8, and secondarily to the transcription factors such as Pax6, Emx2, Coup-TFI, and Sp8 [52,53,54,55,56]. It is of note that these key transcription factors are expressed in a gradient (Figure 2A). Pax6 and Sp8 are expressed in a rostro-lateral^high^ to caudo-medial^low^ manner along the AP axis. Emx2 and Coup-TF1 are expressed inversely to Pax6 and Sp8 [55,57,58,59]. Changes in the expression of these transcription factors affect brain patterning. For example, the overall cortical regions in *Sey/Sey* mice shift to rostral, resulting in a reduction in the motor (M1) and sensory (S1) cortical areas, and conversely an enlargement of the auditory (A1) and visual (V1) areas (Figure 2B) [57,60]. In contrast, the cortical areas in *Emx2* mutant mice shift to caudal, resulting in enlarged M1 and S1 areas and reduced A1 and V1 areas [57,60,61]. *Yac-Pax6* Tg mice, in which Pax6 is overexpressed, show a slight decrease in the S1 area but no major changes in other brain areas (Figure 2C) [62,63]. These results suggest that the proper amount of Pax6 regulates the arealization of the cortex. Furthermore, Pax6 is regulated by Emx2 and Coup-TF1, which determine the caudal area of the cortex (Figure 2B–D) (see [51] and the references therein), indicating that regulatory networks including Pax6 and other transcriptional factors contribute to the AP patterning of the telencephalon.

The expression of the transcription factors is induced by specific secreted molecules (Figure 2D). For example, BMP and Wnt, being released from the dorso-medial telencephalon, increase the expression of Emx2 [52], while Fgf15 from the rostral side induces that of Coup-TF1 [64]. In contrast, Fgf8 from the rostral side induces Sp8 [55,59]. A mathematical analysis of gene regulatory networks using Boolean and computational models has revealed an underpinning expression gradient in the mammalian cortex [65,66]. Although upstream secretary molecules that can induce the expression of Pax6 have not been identified yet, it is of interest that Pax6 controls the responsiveness to secreted molecules such as sonic hedgehog and BMP in the cortex [67]. Future analyses may reveal more about the relationship between Pax6 and secreted molecules.
Figure 2Gradient expression of key transcription factors in cortical patterning. (**A**) Gradient expression patterns of the major transcription factors Pax6, Emx2, Coup-TFI, Sp8, Fgf8, and Emx1 along the anteroposterior (AP) and lateromedial (LM); anterolateral (AL) and posteromedial (PM) axes. Pax6 is highly expressed along the AL axis, and weakly along the PM axis. *Emx2* is expressed inversely to *Pax6*. (**B**,**C**) Summary of loss-of-function (**B**) and gain-of-function (**C**) by the alteration of transcription factor expression in regard to cortical patterning. Analyses of brain formation indicate that *Sey/Sey* mice show a reduction in the M1 and S1 areas, as well as enlargement of the A1 and V1 areas. Embryonic analyses imply that *Emx2* mutants show an inverse relationship to *Sey/Sey* mice. The overexpression of *Emx2* under the control of a Nestin promoter increases the size of the V1 area. In contrast, the overexpression of *Pax6* slightly reduces the size of the S1 area. (**D**) The regulatory network of transcription factors and secreted molecules. The BMP and Wnt gradients increase the expression of Emx2. Fgf15 enhances the expression of Coup-TF1. Fgf8 inhibits the expression of Emx2 and Coup-TF1. TF: transcriptional factor. The graphical diagram has been redrawn from [51,54,55,56,60,61,63,66,68,69,70].
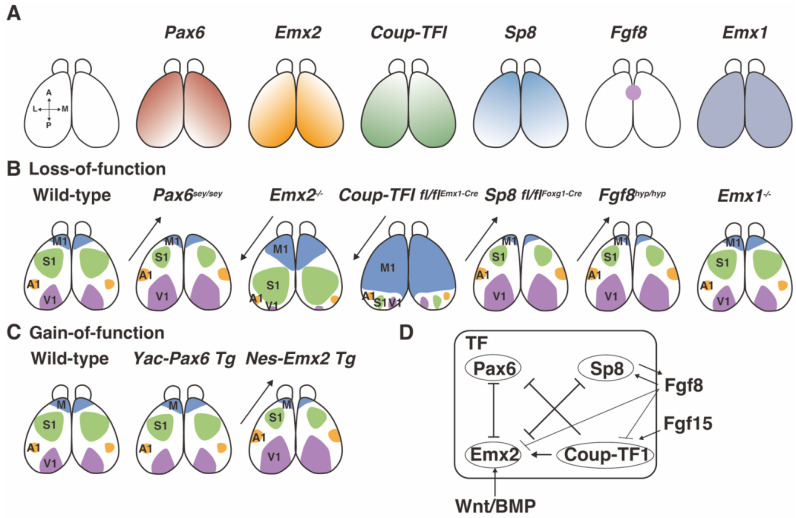


### 2.2. Dorsoventral Axis

Dorsoventrally, the telencephalon is divided into two compartments: the dorsal cortex and the ventral basal ganglia, each of which has specific molecular features [71]. For proper patterning along the DV axis, the coordination of multiple transcription factors is required. In this section, we will review that Pax6 also influences DV patterning in the telencephalon.

The homeobox genes *Pax6* and *Gsh2* are expressed in the cortex and lateral ganglionic eminence (LGE), respectively. It is of note that these two transcriptional factors play complementary roles in the DV patterning of the mammalian telencephalon [72,73]. In *Pax6*-deficient mice, the region expressing *Ascl1* (*Mash1*) and *Dlx*, which is originally localized in the basal ganglia, shifts dorsally, while that expressing *Neurogenin1/2* (*Neurog1/2*), which is originally localized in the dorsal cortex, shifts dorsomedially [74]. In *Gsh2*-deficient mice, conversely, the region expressing *Ascl1/Dlx* and *Neurog1/2* shifts ventrally (Figure 1C) [74,75]. It has been indicated that *Gsh2* is required for repressing *Pax6* and vice versa, and Pax6 is required for maintaining *Neurog1/2*, which means, in turn, Pax6 is responsible for the repression of *Ascl1/Dlx* [74,76]. These results indicate that Pax6 defines the DV boundary region and contributes to DV progenitor identity acquisition.

In addition to the dorsalization of the embryonic telencephalon, our group has found another key transcription factor under the control of Pax6. From microarray analyses using *rSey^2^/rSey^2^* rat embryos, we have identified *Dmrta1* (doublesex and mab-3-related transcription factor-like family A1) as a Pax6 downstream target gene in the rat telencephalon [77]. Dmrta1 is specifically expressed in the dorsal telencephalon and contributes to DV patterning [77]. Dmrta1 overexpression in the rat ventral telencephalon induced the mis-expression of the dorsal marker Neurog2 and repressed that of the ventral marker Ascl1 [77]. These novel pathways, i.e., Pax6 → Dmrta1 → Ngn2-|Ascl1 and/or Pax6 → Dmrta1-|Ascl1, could determine progenitor cell DV identity by repressing ventralization. For more information on the roles of Dmrt family members in brain patterning, see the review in [78]. As described in this section, Pax6 promotes brain development via numerous downstream factors to maintain proper AP and DV patterning in the telencephalon.

## 3. Pax6 in Neurogenesis

During embryonic brain development, neuroepithelial cells initially expand their population by symmetric cell division (Figure 3A, the proliferation phase) [79,80]. Later, the neuroepithelial cells become thinner and longer and are called radial glial (RG) cells. The RG cells divide symmetrically or asymmetrically to self-renew themselves or to produce neurons, respectively (Figure 3A, the neurogenetic phase). After neural differentiation, RG cells produce glial cells, i.e., first astrocytes and then oligodendrocytes, in the cortical primordium (Figure 3A, the gliogenic phase). The neuroepithelial cells and RG cells function as NSCs.

The relative proportion of Pax6-positive VZ in the telencephalon is largest at around embryonic day (E) 12.5 [81], at the time of the transition between the proliferation of NSCs and the induction of neurons (Figure 3B,C) [20,71,82]. Pax6-positive NSCs sequentially differentiate into neurons via neural progenitor cells, so-called intermediate progenitors (IPCs) (see below) [83].

Neurogenesis, i.e., production of neurons, occurs simultaneously with cortical patterning, and these two phenomena are difficult to separate. It is currently believed that neurogenesis continues throughout life in certain brain regions such as the hippocampus dentate gyrus and the subventricular zone of the lateral ventricle [84,85,86,87]. Pax6 is expressed in NSCs at all stages of embryonic, postnatal, and adult neurogenesis (see “Pax6 in neurogenesis” and “Pax6 in Relation to Neurodevelopmental Disorders”) (Figure 3B) [88,89,90]. In addition, Pax6 is expressed in astrocytes [91,92], a type of glial cell sharing some features with NSCs, maintaining a good balance between the maintenance and differentiation of astrocyte progenitors [91]. The determination of the proper neuronal arrangement in the telencephalon proceeds by the following two steps; (i) cell proliferation, and (ii) neural differentiation. In this section, the two different steps regulated by the downstream genes of Pax6 will be reviewed.

### 3.1. Downstream Genes of Pax6 Contribute to Self-Renewal of NSCs

As previously mentioned in the introduction of this paper, NSCs proliferate before neural differentiation proceeds (Figure 3A). The cell cycle consists of a series of intracellular processes, i.e., promoting cell division and producing two daughter cells. Cyclin D2 is essential for the transition from the G1 phase (when the cellular and extracellular environment is checked) to the S phase (DNA synthesis phase). The daughter cells expressing cyclin D2 are maintained in an undifferentiated state with proliferative potential, whereas those that lack cyclin D2 exit the cell cycle and give rise to neurons [93]. Cyclin D2 might be negatively regulated by Pax6, since cyclin D2 expression is increased in the *Sey/Sey* telencephalon at E12.5 [94] and in the cortex at later stages (our unpublished data). These cyclin D2-expressing cells might promote cell cycle re-entry because of an increased S phase population in the *Sey/Sey* telencephalon at E15.5 [95]. Interestingly, the S phase length in *Sey/Sey* mice is reported to be shorter at E12.5 but longer at E15.5 [95]. Cyclin D2 is also expressed in NSCs in the ganglionic eminence, and these cyclin D2-expressing cells gradually migrate toward the dorsal cortex. Based on the information above, the effects of cyclin D2 in the *Sey/Sey* cortex should be carefully examined regarding the embryonic stages and radial versus tangential migration of neurons.

One unique phenomenon that occurs during cell proliferation in the cortex is called interkinetic nuclear migration (INM), originally named as “elevator movement” by Fujita et al. [96,97], in which the nucleus of the neuroepithelial/RG cell goes up and down, from the apical to the basal side within the VZ according to the cell cycle [79,98]. *rSey^2^/Sey^2^* rat cortexes shows ectopic INM, suggesting that Pax6 is involved in its regulation [20]. We also found that Ninein, which is a centrosome protein downstream of Pax6, regulates the dynamics of INM by anchoring to microtubules (Figure 4A–C) [99]. Pax6 and its downstream gene *Ninein* are involved in the regulation of INM, thereby serving to ensure the proliferation of NSCs.

In addition to the cell cycle-related regulation described above, chromatin structures are also key for maintaining NSCs in an undifferentiated state [100,101,102,103]. Pax6 is reported to bind to chromatin remodeling complexes including the RE1 silencing transcription factor (REST) and BRG1/BRM-associated factor (BAF) [104,105,106]. From chromatin immunoprecipitation (ChIP) analyses, it has been shown that the Baf170 subunit, a direct Pax6-interacting protein, recruits the REST-corepressor complex, thereby regulating the binding efficiency of target genes that induce IPCs and early-born neurons [105]. Since REST is expressed in NSCs in the cortical primordia (Figure 4A) [107,108], the Pax6/REST complex is responsible for the regulation of maintenance of NSCs and neural differentiation at the appropriate time.

Another interesting gene downstream of Pax6 is *Fmr1*, a causative gene for fragile X syndrome, one of the neurodevelopmental disorders. A previous ChIP-chip analysis suggested that Pax6 binds to the promoter of the *Fmr1* gene [109]. *Fmr1* encodes an RNA-binding protein, FMRP, which maintains NSCs [110]. A knockdown of *Fmr1* and *Fmr1* knock-out mice resulted in a reduction in the size of the Pax6+ NSC pool due to an NSC-to-IPC cell fate change [110]. A novel finding is that FMRP localized in the basal endfeet of NSCs transports its target mRNAs, leading to the regulation of localized mRNAs (Figure 4A) [111]. Moreover, FMRP-target mRNAs during corticogenesis include various genes related to neurodevelopmental disorders [112]; some of them could regulate the proper maintenance of NSCs and might explain mental retardation, one of the symptoms of fragile X syndrome. Since *Pax6* is considered as an ASD risk gene (see “Pax6 in neurodevelopmental disorders” and [42]), the Pax6–Fmr1 regulation in corticogenesis may provide new insights into the pathogenesis of ASD.

### 3.2. Downstream Genes of Pax6 Contribute to Neural Differentiation

The transition from cell proliferation to neural differentiation of NSCs is governed by the expression of pro-neural genes [80,113,114,115]. Pax6 binds to the enhancer region of a pro-neural gene, *Neurog2*, and directly induces its expression, which in turn downregulates expression of Pax6 itself (Figure 4B) [116,117]. NSCs differentiate into IPCs with the up-regulated expression of Tbr2 (EOMES), a T-domain transcription factor that is induced by *Pax6* [83]. Then, the induction of Tbr2 promotes the differentiation of IPs into postmitotic projection neurons via up-regulation of Tbr1 expression [83]. These facts indicate that a sequential molecular cascade of “Pax6 → Tbr2 → Tbr1” correlates with the cell type change of “RGs → IPs → postmitotic projection neurons” according to differentiation [83]. Another group has reported that *Tbr2* is a downstream gene of *Neurog2* [118]. Pax6 thus regulates multiple downstream genes, i.e., Pax6 → Tbr2 → Tbr1, Pax6 → Neurog2 → Tbr2 → Tbr1, and Pax6 → Neurog2 → Insm1 → Neurod1 → Tbr1 (Figure 4D) [119,120]. Although further fate mapping analyses at the single cell level are needed, the molecular machinery underlying Pax6 downstream cascades may regulate cell fate determination in the proper timing of neural development. It is of note that the transition of transcription factor expression is also recapitulated in adult neurogenesis (Figure 5).

Neurogenesis proceeds through positive and negative regulation of multiple transcription factors. One major signal that regulates many downstream transcriptomic factors is Notch signaling, which is a widely conserved signaling pathway from *Drosophila* to mammals [121]. Notch signaling is well known to determine cell fate choice, such as cell proliferation versus differentiation, during the formation of various tissues including the CNS [122,123,124,125,126]. Although Pax6 is conserved in both the chick and mouse telencephalons, the mechanism of neural differentiation provided by Pax6 differs among species and developmental stages [127]. In chicks and in the early stage of the mouse cortex, Pax6 transiently suppresses Notch signaling and induces neural differentiation (Figure 4D), while in the mid/late stages of the mouse cortex, Pax6 maintains NSCs. Therefore, it is speculated that a spatiotemporal dual function of Pax6 in the mouse cortex could lead to the generation of the sophisticated mammalian brain architecture.

PAX6 is also expressed in human neuroectoderm (NE) tissue [128]. Interestingly, PAX6 is uniformly expressed in the NE cells of human fetuses and those of differentiated cells [128]. However, Pax6 is expressed in restricted mouse brain regions during later development [128]. The human neocortex is enlarged compared to that of the mouse, possibly due to the function of basal RG cells [129]. Pax6 is highly expressed in primate but not mouse basal progenitors. Wong et al. demonstrated that sustained Pax6 expression specifically in BP-producing apical RG cells induces primate-like progenitor cells, indicating that sustained PAX6 expression in basal progenitors could be a key feature of subventricular enlargement in the human brain [130].

PAX6 dysfunction is also involved in the pathogenesis of glioblastoma. Transcriptomic and epigenomic analyses have revealed that PAX6/DLX6 promotes the differentiation of WNT5A-mediated glioblastoma stem cells into endothelial-like cells, which serve as an environmental niche supporting the growth of invasive glioma cells throughout the brain parenchyma [131]. In a controversial study, *PAX6*-knock out human neuronal epithelioma cells display increased proliferation and colony-forming abilities, indicating that PAX6 functions as a tumor suppressor [132]. Actually, there are case reports showing downregulation of *PAX6* in glioma samples (see reviews [133] and the references therein). In this way, the formation/prevention of brain tumors is another example of the multiple functions of PAX6 in cell fate determination.

In this section, we have discussed that Pax6 and its downstream transcriptional factors and signaling molecules are important for neural fate determination, where these transcriptional factors have stage- and cell type-specific regulatory mechanisms. A recent preprint article has reported that a conditional knock-out of *Pax6* ectopically induces GABAergic interneurons, as well as oligodendrocyte precursor cells in the cortex [67]. This might be reasonable because cyclin D2 is upregulated in the *Sey/Sey* mouse cortex ([94,134] and our unpublished data). Taken together, Pax6-regulated cell proliferation and neural differentiation plays pivotal roles in proper timing during brain formation.
Figure 4The function of Pax6 and downstream genes. (**A**) Subcellular gene localization in radial glial (RG) cells. Genes localized in the cell nucleus, cytoplasm, and apical and basal endfeet. The graphical diagram has been redrawn from [135]. (**B**) Downstream genes regulated by Pax6 and their roles. Pax6 regulates *Fabp7*, *Ninein,* and *Lewis X*, which are involved in stem cell self-renewal. Pax6 induces *Ngn2*, *Tbr2,* and *Dmrta1*, which induce neural differentiation. (**C**) The Pax6–Ninein network regulates interkinetic nuclear migration (INM) during cell-cycle progression in neuroepithelial cells. The microtubule cytoskeleton plays an important role in INM, in which the nuclei of neuroepithelial cells move apically during G2 phase and basally during G1 phase. Ninein, downstream of Pax6, anchors microtubules during elevator movements to control the dynamics of INM. The graphical diagram has been redrawn from [99]. (**D**) Pax6–Notch interaction for neurogenic programs in the developing mouse cortex. Pax6-dependent neural differentiation by Notch signaling inhibition generates deeper layer (DL) neurons in the early neurogenic phase. Pax6-dependent self-renewal of the RG cells which give rise to upper layer (UL) neurons occurs in the middle/late neurogenic phase when Notch signaling is absent. In this way, the Pax6–Notch pathway coordinates the balance between self-renewal and neural differentiation. The graphical diagram has been redrawn from [127].
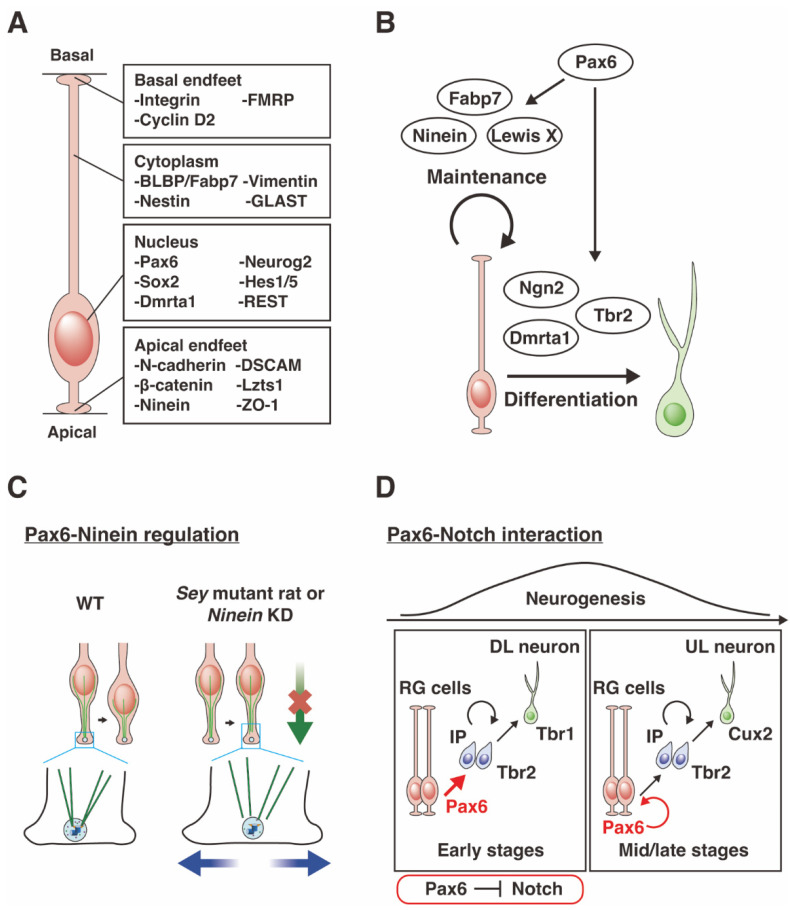

Figure 5Linage and gene expression of NSCs in the adult hippocampal dentate gyrus. Pax6 is expressed in quiescent and active NSCs and intermediate progenitor cells (IPCs). The graphical diagram has been redrawn from [20,88,91,136,137,138,139,140,141,142,143,144].
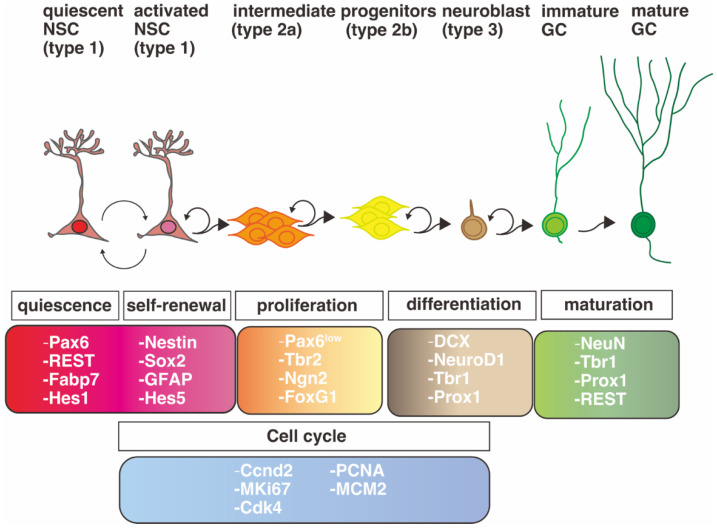


## 4. Pax6 in Relation to Neurodevelopmental Disorders

### 4.1. Human PAX6 Gene Is Related to Neurodevelopmental Disorders

The human *PAX6* gene was discovered during the search for the gene responsible for WAGR (Wilms tumor, aniridia, genital ridge defects, mental retardation) syndrome; patients with the syndrome often show a deletion in chromosome region 11p13 [145]. *PAX6* was originally identified as *AN2* (aniridia type II protein) [2] and found to be a homolog of the causative gene in *Sey/Sey* mice [1], as mentioned above. Another aniridia gene *AN1* was previously mapped to chromosome two, although this mapping was disproven in 1992 [146]. Therefore, *AN2* is now designated *AN1,* and *PAX6* is currently shown as *AN1* in the OMIN database ([147,148], https://omin.org/entry/106210 (accessed on 19 May 2022)).

A causative gene for a kidney disease (Wilms tumor), *WT1*, is located 0.7 Mb away from *PAX6* [149]. Interestingly, a case report has suggested that a 1.6 Mb region containing *PAX6*, *WT1*, and *PRRG4* is responsible for the severe developmental delays and autistic behaviors seen in WAGR syndrome [150]. There is another report showing that WAGR patients sometimes show symptoms of autism in addition to mental retardation [151].

Since aniridia is an obvious congenital disease, patients are often diagnosed genetically. There are many clinical reports showing that aniridia patients with mutations in the *PAX6* gene often exhibit neural phenotypes at the structural and functional level (see [152,153] and the references therein). It has been reported, for example, that patients with a deficiency in the *PAX6* 3′ region containing its enhancer confer ASD and moderate mental retardation, indicating the role of PAX6 in neural phenotypes in addition to aniridia. Our group has also reported 15 single nucleotide polymorphisms within the *PAX6* locus in Japanese autistic patients [154]. Currently, *PAX6* is listed in the Simons Foundation Autism Research Initiative (SFARI) database in the category of “syndromic” [155,156].

Additional evidence for the involvement of the human *PAX6* gene in neurodevelopmental disorders comes from a genome-wide association study (GWAS) of ASD patients [157]. It is of note that the ASD subject group, consisting of both males and females, showed a high odds ratio in the 11p13 region where *PAX6* is localized, but a group consisting of only males did not show a significant odds ratio in the same region. There are not many genetic studies focusing on gender differences in ASD, but a missense mutation of *CTNND2* (δ-catenin) has been identified in female-enriched families containing multiple patients [158]. Since *δ-catenin* is proven to be downstream of *Pax6* in the embryonic mouse [159], it is speculated that *PAX6* and *CTNND2* might be involved in the neuropathology of a severe type of ASD often reported in girls. Our group has found that the reduction in the brain regions of *rSey^2^/+* rats is more severe in the female than the male (Figure 6B,C) [160]. It is reasonable to assume that the functional impairment of Pax6, as well as δ-catenin—both of which are expressed in NSCs in the initial stage of cortical development—may cause a severe reduction in neurons in the cortex, resulting in mental retardation or intellectual disability.

### 4.2. Pax6 Deficient Rodent Models for Neurodevelopmental Disorders

The responsibility of Pax6 in ASD-like phenotypes is modeled in rodents. Cortex-specific Pax6 knock-out (Pax6fl/fl; Emx1-Cre) mice exhibit deficiencies in sensorimotor information integration, as well as both hippocampus-dependent short-term and neocortex-dependent long-term memory recalls [163]. We have identified that *rSey^2^/+* rats show impaired sensorimotor gating, abnormal social interaction, and impaired rearing activity, fear-conditioning, memory, and vocal behavior in pups [164,165]. We have further analyzed the behaviors of *Sey/+* mice and found abnormal vocalization in *Sey/+* pups derived from young fathers and an increase in hyperactivity in those derived from aged fathers [166]. The fact that mice with a single genetic risk factor, *Pax6*, can develop different phenotypes depending on paternal age has alerted basic researchers to the need for considering not only genetic factors but also non-genetic factors in animal models. Researchers had been careful about the age of female mice yet forgot about that of male mice. Another lesson learned here is that non-genetic factors (e.g., paternal age) can mask causative genes in the genetic analysis of neurodevelopmental disorders such as ASD and attention deficit/hyperactivity.

Another *Pax6* mutant strain, *Pax6^Leca2^*, exhibits an impaired retinal structure [167], alteration of circadian clock, and hyperactivity during the light period [168], which is partly consistent with our study in *Sey/+* mice derived from aged fathers [166]. Interestingly, a core circadian clock gene, *Clock,* positively regulates *Pax6* [169], and *Pax6* mutants show an altered expression of circadian clock genes [168]. Our group has also elucidated that Pax6 regulates expression of the Fabp7 gene [170], another candidate associated with circadian rhythm [171] (also involved in the dataset by [172]). Sleep is regulated by circadian rhythm, which is considered as a translationally relevant endpoint in studies of ASD [173]. Therefore, Pax6-circadian clock gene regulation in the brain may influence the pathogenesis of ASD. As supportive evidence, patients with *PAX6* mutation who exhibit aniridia also have a smaller pineal gland, where melatonin is produced, and are often diagnosed with sleep disorders [174]. Analyses using *Pax6* mutant mice suggest that the dysfunction of the eyes might be the direct cause of the alteration of the circadian clock, rather than an impairment of the pineal gland in the brain [168]. Detailed investigation of *Pax6*-circadian clock gene regulation in the eyes and brain will shed further light on the pathogenesis of ASD and comorbid sleep disorders.

## 5. Closing Remark

Pax6 is one of the key molecules from the initial stage of neural development. It regulates brain patterning and the balance between cell proliferation and neural differentiation in an appropriate manner to developmental timing. Multiple context-dependent functions of Pax6 are governed by numerous downstream factors. Patients with mutations in the *PAX6* gene may exhibit symptoms of neurodevelopmental disorders, including ASD. Since *Pax6* mutant rodents exhibit diverse neurodevelopmental phenotypes, they could be used as a tool to elucidate the human pathology of neurodevelopmental disorders. It has also been reported that a Pax6 partner, REST, is expressed from the young to the aged human brain and that the expression level of REST is proportionally related to cognitive function [175]. Thus, the PAX6/REST complex might also be involved in neurodevelopmental and psychiatric disorders, possibly through the regulation of the cell fate decisions of NSCs during embryonic, postnatal, and adult neurogenesis. Further studies on Pax6 and its partner/downstream molecules may lead to the elucidation of the basic molecular mechanisms of neuropsychiatric disorders for future therapeutics.

## Figures and Tables

**Figure 3 ijms-23-06115-f003:**
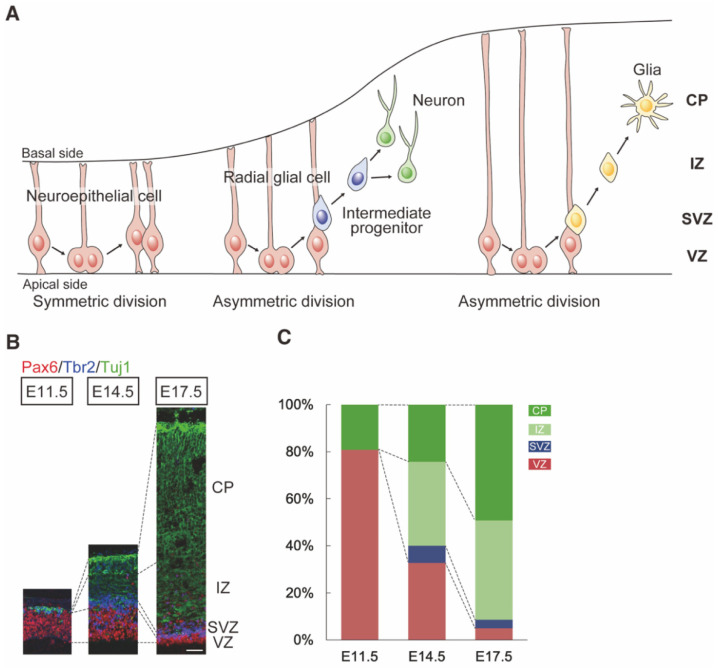
Neural stem cell (NSC) differentiation in the embryonic cortex. (**A**) Neuroepithelial cells undergo symmetrical cell division to increase the population of neural stem cells (NSCs, proliferation phase). When brain development progresses, neuroepithelial cells elongate their processes stretching from the apical surface of the ventricular zone (VZ) to the basal tip (basal endfoot) at the pia surface; these cells are now called radial glial (RG) cells. They undergo asymmetrical cell division and induce intermediate progenitor cells (IPCs), as well as neurons. Intermediate progenitors migrate to the subventricular zone and differentiate into neurons. Subsequently, these neurons migrate towards the basal side. After producing neurons, RG cells produce glia, i.e., astrocytes and oligodendrocytes. The graphical diagram has been redrawn from [80]. (**B**) Expression of Pax6, Tbr2, Tuj1 in the developing cortex at E11.5, E14.5, and E17.5: Pax6 (red), Tbr2 (blue), and Tuj1 (green). These proteins are expressed in the VZ, subventricular zone (SVZ), intermediate zone (IZ), and cortical plate (CP), respectively. Scale bar: 100 μm. (**C**) The transition of the relative volume of brain subdivisions in the developing cortex. The Pax6-positive VZ area gradually narrowed during development.

**Figure 6 ijms-23-06115-f006:**
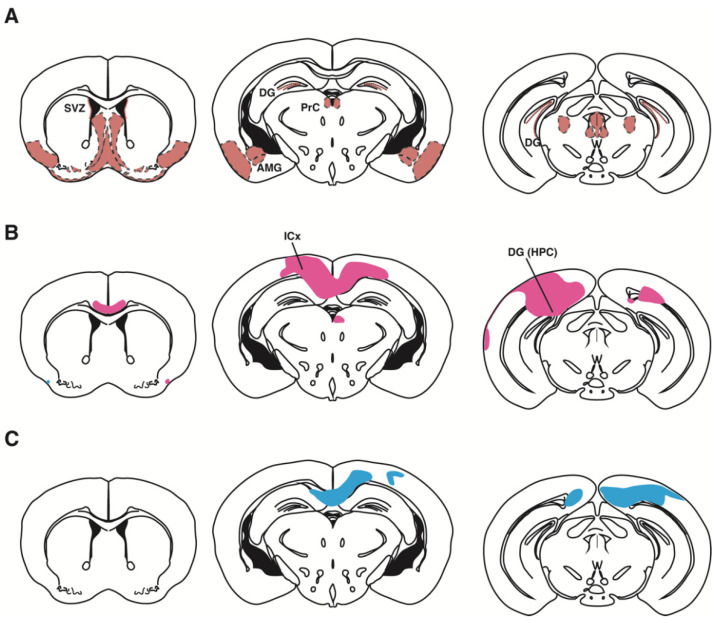
Expression patterns of Pax6 in the mouse brain and the regional volume decrease in the *Pax6* heterozygous mutant (*rSey^2^/+*) rat brain. (**A**) Expression patterns of Pax6 in the adult mouse brain have been redrawn from [25,88,160,161]. The nomenclature and subdivided brain regions are based on previous literature [162]. (**B**,**C**) The sex differences in regional volume decrease in the brain of the *rSey^2^/+* rat compared to the WT using a deformation-based morphometry analysis of MRI data [160]. Pink and blue represent the clusters of regional volume decreases in the brain of female (**B**) and male (**C**) *rSey^2^/+* rats compared to WT rats, respectively. The pink region is larger than the blue region. The graphical diagram has been redrawn from [160]. Abbreviations; AMG: amygdala, DG: dentate gyrus, HPC: hippocampus, ICx: isocortex, PrC: precommissural nucleus, SVZ: subventricular zone.

## Data Availability

Not applicable.

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
