# Peer review of "Thirty Years’ History since the Discovery of Pax6: From Central Nervous System Development to Neurodevelopmental Disorders"

_ijms, 2022, doi:10.3390/ijms23116115_

Round 1

Reviewer 1 Report

The present review by Shohei Ochi and co-workers “30 years history since the discovery of Pax6: from development to disease” is driven by a steady progress over 30 years in our understanding how this transcription factor controls multiple stages of brain development. The Title and Abstract require edits as this review is focused on central nervous system development. For example, text (lines 16-19) can be merged with text in lines 7-8. Pax6 is transcriptional regulator but is would be better to state that it is a sequence-specific DNA binding transcription factor that positively and negatively regulates transcription. Abbreviation for autism spectrum disorder (ASD) should be included in the Abstract. All six figures are well organized and highly informative. If the title talks about disease, why not to discuss important roles of PAX6 in glioblastoma (PMID: 27863244, 29716531 and etc.). Taken together, this review manuscript provides novel summary insights to the field of developmental neuroscience and is within the scope of the International Journal of Molecular Sciencesand special issue on Pax genes/transcription factors.

Additional comments/suggestions:

1) As this is a review article, the list of references cannot be ever “complete”. Nevertheless, earlier comprehensive reviews on eye and pancreas have been ignored (e.g. PMID:22561546, 27126352, and 31325538).

2) Introduction: Hogan et al. PMID: 3250848 was first to show in detail ocular abnormalities in Sey/Sey

3) Introduction: References [7] and [12] are incorrect.

4) Figure 1: The figure shows Pax6 proteins but the title is “Schematic structure of Pax6gene …”. In addition, the next needs to mention Pax6(5a) splice variant and HD-only Pax6, including the role of HD in brain development (PMID:15548580).

5) Introduction (lines 56-60): Expression of Pax6 in periocular mesenchyme/neural crest and olfactory placodes is ignored.

6) Section 2 (Lines 75-77). Recent single cell RNA-seq data show Pax6 expression at E7.5 of mouse development (PMID: 31391582).

7) Section 2: Why not to discuss the role of PAX6 in human neuroectoderm (PMID: 20621053) and differences between mouse and human cortical development (Wong et al. 2015, PMID: 26252244).

8) Figure 2D: FGF15 or Fgf8? Keep consistent all protein names.

9) Section 3.1 (lines 228/233): Reference [96] is only on Pax6 and REST. Maybe add details such as that Baf170 subunit, a direct Pax6 interacting protein, recruits REST-corepressor complex shown by ChIP studies. Explain better roles of Pax6 in transcriptional repression.

10) Neuroanatomical abnormalities in Aniridia patients and mouse models have been analyzed in a series of papers from the J.D. Lauderdale lab.

Author Response

Reviewer’s comment
1. As this is a review article, the list of references cannot be ever “complete”. Nevertheless, earlier comprehensive reviews on eye and pancreas have been ignored (e.g. PMID:22561546, 27126352, and 31325538).

Our response

We thank this reviewer for reminding important comprehesive reviews about Pax6 function on the eye and pancreas that we have missed. We now cite the previous papers as follows (p.2, line 59-60):
“Pax6 is also involved in development of other tissues such as the pancreas, pituitary gland, and even in the testis [21, 38-40] (reviewed in [13, 20, 37, 41-44]).”

During this revision process, we have also noticed that another important review (Manuel et al., 2015) was missing, which has also been added in the text as the reference [44] (p2, line 60).

2. Introduction: Hogan et al. PMID: 3250848 was first to show in detail ocular abnormalities in Sey/Sey.

Our response

We thank this reviewer for reminding an original article about detailed ocular abnormalities in Sey/Sey. We now cite the paper as [11] in p.1, line 41-43, also reflecting the comment from reviewer#2.

“The Pax6 gene was also identified in spontaneous mutant mice (Sey/Sey) and rats (rSey2/rSey2) showing similar ocular phenotypes [11-13]”

3. Introduction: References [7] and [12] are incorrect.
Our response

We thank the reviewer for bringing this point to our attention. We changed the references as follows (p.1-p.2, line 43-45):
“The gene responsible for the eyeless mutant in Drosophila was found to be a Pax6 homolog [14, Quiring et al., 1994]. Ectopic expression of eyeless induced the formation of compound eyes outside the head region [7, Halder et al., 1995]”

4 Figure 1: The figure shows Pax6 proteins but the title is “Schematic structure of Pax6gene …”. In addition, the next needs to mention Pax6(5a) splice variant and HD-only Pax6, including the role of HD in brain development (PMID:15548580).

Our response

We thank this reviewer for reminding an important article about Pax6(5a) splice variant and HD-only Pax6. We now modify the text in p1, line 34-36, and the title and text of Figure 1 (p3, line 65-67, 69-71 and 76-77) and cite the previous paper as follows. We have also simplified Figure 1A by excluding the construct of Pax6ΔPD.

“It has been reported that PD is necessary for the regulation of embryonic neurogenesis, in which a mutant lacking HD of Pax6 shows only subtle defects [9]. Therefore, PD exhibits a key role for the brain development [9].”

“Figure 1. Schematic structure of Pax6 protein and cortical phenotypes in the Pax6 spontaneous homozygous mutant (Sey/Sey) and Pax6 splice variant (Pax6(5a)) mice. (A) Schematic structure of Pax6 proteins in the wild-type (WT), Sey/Sey and Pax6(5a) mice.”

“The canocial PD binds via its N-terminal PAI domain to DNA, while the insertion of the 14 amino acids from the exon 5a into the PAI domain [PD(5a)] [45, Epstein et al., 1994].”

“The graphical diagram has been revised from [9, 20, 46].”

5 Introduction (lines 56-60): Expression of Pax6 in periocular mesenchyme/neural crest and olfactory placodes is ignored.

Our response

We thank this reviewer’s comment on expression of Pax6 in tissues outside the CNS. Actually, Pax6 is not expressed in the periocular mesenchyme and neural crest. We have originally published a paper regarding the role of Pax6 on the migration of neurao crest cells derived from the midbrain in the rat embryo (Matsuo, Osumi-Yamashita et al., Nat Genet, 1993). Later we have shown that Pax6, which is expressed in the frontonasal epithelium, plays a non-cell autonomous role on the migration of midbrain crest cells that contribute to the periocular mesenchyme (Nagase et al., Dev Growth Differ, 2001; Nagase et al., J Anat, 2003).

We now add a key review article on the role of Pax6 in eye development ([37] Cvekl and Callaerts, 1999) as follows (p.2, line 57-58):

“Outside the CNS, PAX6/Pax6 is expressed in the lens [20, 30, 31], corneal epithelium [32, 33], retinal neuroepithelium [34, 35], and olfactory placodes/epithelium [25, 36, 37].

6 Section 2 (Lines 75-77). Recent single cell RNA-seq data show Pax6 expression at E7.5 of mouse development (PMID: 31391582).

Our response

We thank this reviewer for reminding an important article about recent single cell RNA-seq data showing Pax6 expression at the neuroectoderm at E7.5.We have changed the description and cited the papers as follows (p.4, line 81-84):

“As mentioned above, Pax6 is considered to be expressed from the initial developmental stage of the CNS, when it plays key roles in regionalization of the neuroectoderm and neural tube, i.e., the primordium of the CNS [47]. Each region of the CNS is specialized before and during neurogenesis [48]

7 Section 2: Why not to discuss the role of PAX6 in human neuroectoderm (PMID: 20621053) and differences between mouse and human cortical development (Wong et al. 2015, PMID: 26252244).

Our response

We thank this reviewer for suggesting the discussion on the the role of PAX6 in human CNS development by reminding important articles. We now cite the previous papers as follows (p.9, line 286-294):

“PAX6 is also expressed in the human neuroectoderm (NE) tissue [128]. Interestingly, PAX6 is uniformly expressed in NE cells of human fetuses and those differentiated cells [128]. However, Pax6 is expressed in restricted mouse brain regions at later development [128]. Human neocortex is enlarged compared to that of the mouse, possibly due to the function of basal RG cells [129]. Pax6 is highly expressed in primate basal progenitors but not in the mouse ones. Wong et al., demonstrate that sustained Pax6 expression specifically in BP-producing apical RG cells induces primate-like progenitor cells, indicating that sustained PAX6 expression in basal progenitor could be a key feature of subventricular enlargement in human brain [130].”

8 Figure 2D: FGF15 or Fgf8? Keep consistent all protein names.

Our response

We appreciated the reviewer’s comment. We changed the protain names to keep the consistency in Figure 2D (p.5).

9 Section 3.1 (lines 228/233): Reference [96] is only on Pax6 and REST. Maybe add details such as that Baf170 subunit, a direct Pax6 interacting protein, recruits REST-corepressor complex shown by ChIP studies. Explain better roles of Pax6 in transcriptional repression.

Our response

We appreciated the reviewer’s comments. We changed the text as follows (p.8, line 234-239).

“Pax6 is reported to bind to chromatin remodeling complexes including RE1 Silencing Transcription Factor (REST) and BRG1/BRM-Associated Factor (BAF) [104-106]. From chromatin immunoprecipitation analyses, it is shown that Baf170 subunit, a direct Pax6 interacting protein, recruits REST-corepressor complex shown by ChIP studies, thereby regulating the binding efficiency of target genes that induce IPCs and early born neurons [105].”

10 Neuroanatomical abnormalities in Aniridia patients and mouse models have been analyzed in a series of papers from the J.D. Lauderdale lab.

Our response

We thank this reviewer for reminding the important articles about aniridia patients and mouse models. We now cite a review paper as follows (p.12, line 351-353):

“There are many clinical reports showing that aniridia patients with mutations in the PAX6 gene often exhibit neural phenotypes in structure and functional level (see [152, 153] and references therein).”

11 The Title and Abstract require edits as this review is focused on central nervous system development.

Our response

We appreciated the reviewer’s critical comment. We edited the title and abstract the as follows (p.1, line 2-3; p.1, line 7-9).

“30 years history since the discovery of Pax6: from the central nervous system development to neurodevelopmental disorders”

“Pax6 is a sequence-specific DNA binding transcription factor that positively and negatively regulates transcription and is expressed in multiple cell types in the developing and adult central nervous system (CNS)”.

12 For example, text (lines 16-19) can be merged with text in lines 7-8.

Our response

We appreciated the reviewer’s comment, and merged the sentences in line 7-8 and in line 16-19 as follows (p.1, line 15-18).

Pax6 gene was originally identified in the mouse and human 30 years ago via genetical analyses of the eye phenotypes. The human PAX6 gene was discovered in patients who suffer from WAGR syndrome (i.e., Wilms tumor, aniridia, genital ridge defects, mental retardation).”

13 Pax6 is transcriptional regulator but is would be better to state that it is a sequence-specific DNA binding transcription factor that positively and negatively regulates transcription.

Our response

We appreciated the reviewer’s comment. We have already reflected the comments regarding another comment above (p.1, line 7-9).

“Pax6 is a sequence-specific DNA binding transcription factor that positively and negatively regulates transcription and is expressed in multiple cell types in the developing and adult central nervous system (CNS).”

14 Abbreviation for autism spectrum disorder (ASD) should be included in the Abstract.

Our response

We included the explanation of the abbreviation in the text as follows (p.1, line 18-19).

“Mutations of human PAX6 gene have also been reported to be associated with autism spectrum disorder (ASD) and intellectual disability.”

15 If the title talks about disease, why not to discuss important roles of PAX6 in glioblastoma (PMID: 27863244, 29716531 and etc.).

Our response

We thank this reviewer for reminding important studies about the role of PAX6 in glioblastoma. As mentioned above, we are focusing on neurodevelopmental disorders, but would like to briefly mention this issue as follows (p.9 line 295-304):

“PAX6 dysfunction is also involved in the pathogenesis of glioblastoma. Transcriptomic and epigenomic analyses have revealed that PAX6/DLX6 promotes differentiation of WNT5A-mediated glioblastoma stem cells into endothelial like cells, which serve as an environmental niche supporting the growth of invasive glioma cells throughout the brain parenchyma [131]. As a controversy study, PAX6-knock out human neuronal epithelioma cells have increased proliferation and colony forming abilities, indicating PAX6 functions as a tumor suppressor [132]. Actually, there are various case reports showing downregulation of PAX6 in glioma samples (see review [133] and references therin). In this way, formation/prevention of brain tumors is another example of multiple functions of PAX6 in cell fate determination.”

Reviewer 2 Report

The manuscript "30 years history since the discovery of Pax6: from development to disease" is a review, written by Ochi S, Manabe S, Kikkawa T and Osumi N. It describes the role of Pax6 transcription factor in brain development. After introductory part it describes the role of Pax6 in cortical patterning and different roles in neurogenesis. At the end, the role of Pax6 in human neurodevelopment disorders is described.

The manuscript is well written, correctly designed, oriented on the narrow field of neural development under Pax6 regulation and illustrated with several schemes. The references cited cover both, the history of Pax6 and up-to date data.

However, there are lots of specific comments:

line 16: The human PAX6 gene has been discovered from patients – sentence reconstruction

line 27: ..as a gene

line 35: Pax 6 was first paid attention – sentence reconstruction

line 40: in spontaneous mutant rats – sentence reconstruction

line 49: neural stem cells are located

line 52: at the rostral most position – sentence reconstruction

line 56: PAX6 or Pax6

line 82: diverse functionally and morphologically – sentence reconstruction

line 100: that determine

line 106: some proteins are written with small and others with capital letters (FGF)

line 110: it

line 112: of new insight in the relationship – sentence reconstruction

line 117: AL and PM axis

line 119: which caused... – sentence reconstruction

line 122: mutants show...relationship to

line 132: coordinating – coordination

line 147: Pax6

line 167: NCS mentioned for the first time,  explanation of the abbreviation

line 168: the same for VZ

line 195: blue

line 203: sentence reconstruction

lines 223, 238, 241, 259: sentences reconstruction

line 245: ASD explanation - line 322 can be omitted

line 287: explanation of RG

line 310: related to

line 316: sentence not clear, is AN2 PAX6 or not

line 329: our group has

line 333: sentence reconstruction

line 335: group consisting of both, males and females

line 339: what are multiplex families

line 350: literature, are

line 353: blue; regional

line 362: as well as both,

Author Response

Responses to comments from reviewer #2

Reviewer’s comments

The manuscript is well written, correctly designed, oriented on the narrow field of neural development under Pax6 regulation and illustrated with several schemes. The references cited cover both, the history of Pax6 and up-to date data.However, there are lots of specific comments:

1 line 16: The human PAX6 gene has been discovered from patients – sentence reconstruction

Our response

We are grateful for many thoughtful comments from the reviewer to improve our manuscript. We rewrote the text in p.1, line 16-18 as follows.

“The human PAX6 gene was discovered in patients who suffer from WAGR syndrome (i.e., Wilms tumor, aniridia, genital ridge defects, mental retardation).”

2 line 27: ..as a gen

Our response

We changed the text in p.1, line 27-28 as follows. We have also reflected the comment from Reviewer#1.

“It has been 30 years since Pax6/PAX6 was identified as a gene responsible for congenital anomalies of the eye in mouse and human [1, 2].”

3 line 35: Pax 6 was first paid attention – sentence reconstruction

Our response

We rewrote the text in p.1, line 37 as follows.

“Pax6 first garnered attention due to its role in eye development.”

4 line 40: in spontaneous mutant rats – sentence reconstruction

Our response

We rewrote the text in p.1, line 41-43 as follows. We have also reflected the comment from Reviewer#1.
“The Pax6 gene was also identified in spontaneous mutant mice (Sey/Sey) and rats (rSey2/rSey2) showing similar ocular phenotypes [11-13].”

5 line 49: neural stem cells are located

Our response

We changed the text in p.2, line 50-52 as follows.

“At early stages, Pax6 is expressed in the ventricular zone (VZ), where neural stem cells (NSCs) are located (see the part of “Pax6 in neurogenesis”) [23, 24].”

6 line 52: at the rostral most position – sentence reconstruction

Our response

We rewrote the text in p.2, line 54-55 as follows.

rSey2/rSey2 rats lack the olfactory bulb, yet they have an olfactory bulb-like structure at the lateral position in the neocortex [20].”

7 line 56: PAX6 or Pax6

Our response

We changed the text in p.2, line 57-58 as follows, also reflecting the comment from Reviewer#1.
“Outside the CNS, PAX6/Pax6 is expressed in the lens [20, 30, 31], corneal epithelium [32, 33], retinal neuroepithelium [34, 35], and olfactory placodes/epithelium [25, 36, 37].”

8 line 82: diverse functionally and morphologically – sentence reconstruction

Our response

We rewrote the text in p.4, line 90-91 as follows.

“The telencephalon is a highly regionalized organ, that is functionally and morphologically diverse.”

9 line 100: that determine

Our response

We changed the text in p.4, line 107-110 as follows.

“Furthermore, Pax6 is regulated by Emx2 and Coup-TF1 that determine the caudal area of the cortex (Fig. 2B-D) ([51] and references therein), indicating that regulatory networks, including Pax6 and other transcriptional factors, contribute to the AP patterning in the telencephalon.”

10 line 106: some proteins are written with small and others with capital letters (FGF)

Our response

We changed the text in p.4, line 112-114 as follows.
“For example, BMP and Wnt, being released from the dorso-medial telencephalon, increase expression of Emx2 [52], while Fgf15 from the rostral side induces that of Coup-TF1 [63]. In contrast, Fgf8 from the rostral side induces Sp8 [55, 59].”

11 line 110: it

Our response

We change the upeercase letter to the lowercase letter in p.4, line 116-119 as follows.

“Although upstream secretary molecules that can induce expression of Pax6 have not been identified yet, it is of interest that Pax6 controls the responsiveness to secreted molecules such as sonic hedgehog and BMP in the cortex [67].”

12 line 112: of new insight in the relationship – sentence reconstruction

Our response

We rewote the text in p.4, line 119-120 as follows.

“Future analyses may reveal more about the relationship between Pax6 and secreted molecules.”

13 line 117: AL and PM axis

Our response

We changed the words in p.5, line 122-125 as follows.

“(A) Gradient expression patterns of the major transcription factors Pax6, Emx2, Coup-TFI, Sp8, Fgf8, Emx1, along the anteroposterior (AP) and lateromedial (LM); anterolateral (AL) and posteromedial (PM) axes. Pax6 is highly expressed along the AL axis, and weakly along the PM axis.”

14 line 119: which caused... – sentence reconstruction

Our response
We rewrote the text in p.5, line 126-127 as follows.

“(B, C) Summary of loss-of-function (B) and gain-of-function (C) by the alteration of transcription factor expression in regard with cortical patterning.”

15 line 122: mutants show...relationship to

Our response

We changed the text in p.5, line 129-130 as follows.

“The embryonic analyses imply that the Emx2 mutants show an inverse relationship to Sey/Sey mice.”

16 line 132: coordinating – coordination

Our response

We changed the text in p.5, line 139-141 as follows.

“For proper patterning along the DV axis, coordination of multiple transcription factors is required.”

17 line 147: Pax6

Our response

We changed the word in p.6, line 154-155 as follows.

“In addition to the dorsalization of the embryonic telencephalon, our group have found another key transcription factor under the control of Pax6.”

18 line 167: NCS mentioned for the first time,  explanation of the abbreviation

19 line 168: the same for VZ

Our response

An appreviation “NSC” and “VZ” have been mentioned for the first time in p.2, line 50-52 as follows.
“At early stages, Pax6 is expressed in the ventricular zone (VZ), where neural stem cells (NSCs) are located (see the part of “Pax6 in neurogenesis”) [23, 24].”

20 line 195: blue

Our response

We changed the text in p.7, line 201-202 as follows.
“(B) Expression of Pax6, Tbr2, Tuj1 in the developing cortex at E11.5, E14.5 and E17.5. Pax6 (red), Tbr2 (blue) and Tuj1 (green).”

21 line 203: sentence reconstruction

Our response

We rewrote the text in p.8, line 208-210 as follows.

 “As previously mentioned in “1. Introduction”, NSCs proliferate before neural differentiation proceeds (Fig. 3A). The cell cycle consists of a series of intracellular processes, i.e., promoting cell division and producing two daughter cells.”

22 lines 223: sentences reconstruction

Our response

We rewrote the text in p.8, line 228-230 as follows.

“We also found that Ninein, which is a cetrosome protein and downstream to Pax6, regulates the dynamics of INM by anchoring to microtubules (Fig. 4A-C) [99].”

23 lines 238: sentences reconstruction

Our response

We rewrote the text in p.8, line 245-246 as follows.

Fmr1 encodes an RNA-binding protein, FMRP, which maintains NSCs [110].”

24 lines 241: sentences reconstruction

Our response

We rewrote the text in p.8, line 248-249.

“A novel finding is that FMRP localized in the basal endfeet of NSCs transports its target mRNAs, leading to the regulation of localized mRNAs (Fig. 4A) [111].”

25 line 245: ASD explanation - line 322 can be omitted
Our response

We deleted the corresponding text and reworte the text in p.8, line 253-255 and p.9, line 348-349 as follows.

“Since Pax6 is considered as an ASD risk gene (see “Pax6 in neurodevelopmental disorders” and [42]), the Pax6-Fmr1 regulation in corticogenesis may provide new insights into the pathogenesis of ASD.”

“There is another report showing that WAGR patients sometimes show symptoms of autism in addition to mental retardation [151].”

26 lines 259: sentences reconstruction

Our response

We reworte the text in p.9 line 266-269.

“Another group has reported that Tbr2 is a downstream gene of Neurog2 [117]. Pax6 thus regulates multiple downstream genes, i.e., Pax6→Tbr2→Tbr1, Pax6→Neurog2→Tbr2→Tbr1, and Pax6→Neurog2→Insm1→Neurod1→Tbr1 (Fig. 4D) [119, 120].

27 line 287: explanation of RG

Our response

We changed the text in p.10, line 313-314 as follows.

(A) Subcellular gene localization in radial glial (RG) cells.”

28 line 310: related to

Our response

We changed the word in p.12, line 336 as follows.

“4.1. Human PAX6 gene is related to neurodevelopmental disorders”

29 line 316: sentence not clear, is AN2 PAX6 or not

Our response

The history of renaming AN1 and AN2 is complicated. We rewrote the text in p.11-12, line 340-344 as follows.
PAX6 was originally identified as AN2 (Aniridia type II protein) [2] and found to be a homolog to the causative gene in Sey/Sey mice [1], as mentioned above. Another aniridia gene AN1 was previously mapped to chromosome 2, although this mapping was disproven in 1992 [146]. Therefore, AN2 is now designated AN1, and PAX6 is currently shown as AN1 in OMIN database ([147, 148], https://omin.org/entry/106210).

30 line 329: our group has

Our response

We changed the word in p.12, line 355-357 as follows, also reflecting the comment from Reviewer#3.

“Our group has also reported 15 single nucleotide polymorphisms in Japanese autistic patients within the PAX6 locus [154].”

31 line 333: sentence reconstruction

Our response

We reworte the text in p.12, line 359-361 as follows.

“Additional evidence for the involvement of the human PAX6 gene in neurodevelopmental disorders is comes from genome wide association study (GWAS) of ASD patients [157].”

32 line 335: group consisting of both, males and females

Our response

We changed the text in p.12, line 361-363 as follows.

“It is of note that the ASD subject group, consisting of both males and females, shows a high odds ratio in 11p13 region, where PAX6 is localized, but those consisting of only males does not show significant odds ratio in the same region.”

33 line 339: what are multiplex families

Our response

“Multiplex family” is a technical term in the genetics. Considering the wide range of readers, we changed the text in p.12, line 363-366 as follows.

“There are not so many genetic studies focusing on gender difference in ASD, but a missense mutation of CTNND2 (δ-catenin) has been identified in female-enriched families containing multiple patients [158].”

34 line 350: literature, are

Our response

We changed the legend for Fig. 6 as below (p13, line 376-378):

“(A) Expression patterns of Pax6 in the adult mouse brain are based on literatures [25, 88, 160, 161]. The nomenclature and subdevided brain regions are based on a previous literature [162].”

35 line 353: blue; regional

Our response

We changed the typos in p.12, line 380-381 as follows.

“Pink and blue represent clusters of regional volume decreases in the brain of female (B) and male (C) rSey2/+ rats compared to WT rats, respectively.”

36 line 362: as well as both,

Our response

We appreciated the reviewer’s comment, and changed the text in p.13, line 387-390 as follows.

“Cortex-specific Pax6 knock-out (Pax6fl/fl; Emx1-Cre) mice exhibit deficiencies in sensorimotor information integration, as well as both hippocampus-dependent short-term and neocortex-dependent long-term memory recalls [163].”

Reviewer 3 Report

This is a comprehensive review on the fundamental role of a PAX6 gene in the differentiation of neural cells in mice and human,  contribution  to self-renual of neural stem-like cells and neurodevelopmental disorders. The attraction of the interestingly written history  of the PAX6 discovery  is increased due to personal professional contribution of the key author of review to the first PAX6 studies  published as early as  in 1991-1993.  The strength of the review is further improved by reviewing the latest studies of the group and other authors describing the role of PAX6 in autism spectrum disorders and  other psychiatric disorders  via regulating cell fate decision by neural stem-like cells  at different stages of neurogenesis. Informative discussion on control of PAX6 gene of downstream genes to develop neuropathological sex-related patterns related to autism may provide a push for future studies.

Text requires minor  spell check (line 384 -mutations, not "mutantations"; line 330: polymorphisms, instead of "morphisms")

Author Response

Responses to comments from reviewer #3

Reviewer’s comments

  1. Text requires minor spell check (line 384 -mutations, not "mutantations")

Our response

We are grateful for the reviewer’s comments to improve our manuscript. Within the entire text, we checked spell/grammar and made various corrections, which are shown as the history of the revised file. For example, we changed the word in p.14, line 410-413 as follows.

“As supportive evidence, patients with PAX6 mutation, who exhibit aniridia, also have a smaller pineal gland, where melatonin is produced, and are diagnosed with sleep disorders [174].”

2. Text requires minor spell check (line 330: polymorphisms, instead of "morphisms").

Our response

We changed the word in p.12, line 355-357 as follows, also reflecting the comment from Reviewer#2.

“Our group has also reported 15 single nucleotide polymorphisms within the PAX6 locus in Japanese autistic patients [154].”